# LSH-SAMPLING BREAKS THE COMPUTATIONAL CHICKEN-AND-EGG LOOP IN ADAPTIVE STOCHASTIC GRADIENT ESTIMATION

**Beidi Chen, Yingchen Xu & Anshumali Shrivastava**
Department of Computer Science
Rice University
Houston, TX 77005, USA
{beidi.chen,yx26,anshumali}@rice.edu

## ABSTRACT

Stochastic Gradient Descent or SGD is the most popular optimization algorithm for large-scale problems. SGD estimates the gradient by uniform sampling with sample size one. There have been several other works that suggest faster epoch wise convergence by using weighted non-uniform sampling for better gradient estimates. Unfortunately, the per-iteration cost of maintaining this adaptive distribution for gradient estimation is more than calculating the full gradient. As a result, the false impression of faster convergence in iterations leads to slower convergence in time, which we call a chicken-and-egg loop. In this paper, we break this barrier by providing the first demonstration of a sampling scheme, which leads to superior gradient estimation, while keeping the sampling cost per iteration similar to that of the uniform sampling. Such an algorithm is possible due to the sampling view of Locality Sensitive Hashing (LSH), which came to light recently. As a consequence of superior and fast estimation, we reduce the running time of all existing gradient descent algorithms. We demonstrate the benefits of our proposal on both SGD and AdaGrad.

## 1 MOTIVATION

Stochastic gradient descent or commonly known as SGD is the most popular choice of optimization algorithm in large-scale setting for its computational efficiency. A typical interest in machine learning is to minimize the average loss function $f$ over the training data, with respect to the parameters $\theta$, i.e., the objective function of interest is

$$\theta^* = \arg\min_{\theta} F(\theta) = \arg\min_{\theta} \frac{1}{N} \sum_{i=1}^{N} f(x_i, \theta). \tag{1}$$

Throughout the paper, our training data $D = \{x_i, \ y_i\}_{i=1}^{N}$ will have $N$ instances with $d$ dimensional features $x_i \in \mathbb{R}^d$ and labels $y_i$. The labels can be continuous real valued for regression problems. For classification problem, they will take value in a discrete set, i.e., $y_i \in \{1, 2, \cdots, K\}$. Typically, the function $f$ is a convex function. The least squares $f(x_i, \theta) = (\theta \cdot x_i - y_i)^2$, used in regression setting is a classical example of $f$.

SGD (Bottou, 2010) samples an instance $x_j$ uniformly from $N$ instances, and performs the gradient descent update:

$$\theta_t = \theta_{t-1} - \eta^t \nabla f(x_j, \theta_{t-1}), \tag{2}$$

where $\eta^t$ is the step size at the $t^{th}$ iteration. The gradient $\nabla f(x_j, \theta_{t-1})$ is only evaluated on $x_j$, using the current $\theta_{t-1}$.

It should be noted that a full gradient of the objective is given by the average $\frac{1}{N}\sum_{i=1}^{N}\nabla f(x_i, \theta_{t-1})$. Thus, a uniformly sampled gradient $\nabla f(x_j, \theta_{t-1})$ is an unbiased estimator of the full gradient, i.e.,

$$\mathbb{E}(\nabla f(x_j, \theta_{t-1})) = \frac{1}{N}\sum_{i=1}^{N}\nabla f(x_i, \theta_{t-1}). \tag{3}$$

This is the key reason why, despite only using one sample, SGD still converges to the local minima, analogously to full gradient descent, provided $\eta^t$ is chosen properly (Robbins & Monro, 1951; Bottou, 2010).

However, it is known that the convergence rate of SGD is slower than that of the full gradient descent (Shamir & Zhang, 2013). Nevertheless, the cost of computing the full gradient requires $O(N)$ evaluations of $\nabla f$ compared to just $O(1)$ evaluation in SGD. Thus, with the cost of one epoch of full gradient descent, SGD can perform $O(N)$ epochs, which overcompensates the slow convergence. Therefore, despite slow convergence rates, SGD is almost always the chosen algorithm in large-scale settings as the calculation of the full gradient in every epoch is prohibitively slow. Further improving SGD is still an active area of research. Any such improvement will directly speed up almost all the state-of-the-art algorithms in machine learning.

The slower convergence of SGD is expected due to the poor estimation of the gradient (the average) by only sampling a single instance uniformly. Clearly, the variance of the one sample estimator is high. As a result, there have been several efforts in finding sampling strategies for better estimation of the gradients (Zhao & Zhang, 2014; Needell et al., 2014; Zhao & Zhang, 2015; Alain et al., 2015). The key idea behind these methods is to replace the uniform distribution with a weighted distribution which leads tp a lower variance.

However, with all adaptive sampling methods for SGD, whenever the parameters and the gradients change, the weighted distribution has to change. Unfortunately, as argued in (Gopal, 2016), all of these methods suffer from what we call the chicken-and-egg loop – adaptive sampling improves stochastic estimation but maintaining the required adaptive distribution will cost $O(N)$ per iteration, which is also the cost of computing the full gradient exactly.

To the best of our knowledge, there does not exist any generic sampling scheme for adaptive gradient estimation, where the cost of maintaining and updating the distribution, per iteration, is $O(1)$ which is comparable to SGD. Our work provides first such sampling scheme utilizing the recent advances in sampling and unbiased estimation using Locality Sensitive Hashing (Spring & Shrivastava, 2017).

## 1.1 ADAPTIVE SAMPLING FOR SGD

For non-uniform sampling, we can sample each $x_i$ with an associated weight $w_i$. These $w_i$'s can be tuned to minimize the variance. It was first shown in (Alain et al., 2015), that sampling $x_i$ with probability in proportion to the $L_2$ norm of the gradient, i.e. $||\nabla f(x_i, \theta_{t-1})||_2$, leads to the optimal distribution that minimizes the variance. However, sampling $x_i$ with probability in proportion to $w_i = ||\nabla f(x_i, \theta_{t-1})||_2$, requires first computing all the $w_i$'s, which change in every iteration because $\theta_{t-1}$ gets updated. Therefore, maintaining the values of $w_i$'s is even costlier than computing the full gradient. (Gopal, 2016) proposed to mitigate this overhead partially by exploiting additional side information such as the cluster structure of the data. Prior to the realization of optimal variance distribution, (Zhao & Zhang, 2014) and (Needell et al., 2014) proposed to sample a training instance with a probability proportional to the Lipschitz constant of the function $f(x_i, \theta_{t-1})$ or $\nabla f(x_i, \theta_{t-1})$ respectively. Again, as argued, in (Gopal, 2016), the cost of maintaining the distribution is prohibitive.

It is worth mentioning that before these works, a similar idea was used in designing importance sampling-based low-rank matrix approximation algorithms. The resulting sampling methods, known as leverage score sampling, are again proportional to the squared Euclidean norms of rows and columns of the underlying matrix (Drineas et al., 2012).

**The Chicken-and-Egg Loop:** In summary, to speed up the convergence of stochastic gradient descent, we need non-uniform sampling for better estimates (low variance) of the full gradient. Any interesting non-uniform sampling is dependent on the data and the parameter $\theta_t$ which changes in every iteration. Thus, maintaining the non-uniform distribution for estimation requires $O(N)$ computations to calculate the weights $w_i$, which is the same cost computing it exactly. It is not even

clear that there exists any sweet and adaptive distribution which breaks this computational chicken-and-egg loop. We provide the first affirmative answer by giving an unusual distribution which is derived from probabilistic indexing based on locality sensitive hashing.

**Our Contributions:** In this work, we propose a novel LSH-based samplers, that breaks the afore-mentioned chicken-and-egg loop. Our algorithm, which we call LSD (**L**SH **S**ampled **S**tochastic gradient **D**escent), are generated via hash lookups which have $O(1)$ cost. Moreover, the probability of selecting $x_i$ is provably adaptive. Therefore, the current gradient estimates have lower variance, compared to a single sample SGD, while the computational complexity of sampling is constant and of the order of SGD sampling cost. Furthermore, we demonstrate that LSD can be utilized to speed up any existing gradient-based optimization algorithm such as AdaGrad (Duchi et al., 2011).

As a direct consequence, we obtain a generic and efficient gradient descent algorithm which converges significantly faster than SGD, both in terms of epochs as well as running time. It should be noted that rapid epoch wise convergence alone does not imply computational efficiency. For instances, Newtons method converges faster, epoch wise, than any first-order gradient descent, but it is prohibitively slow in practice. The wall clock time or the amount of floating point operations performed to reach convergence should be the metric of consideration for useful conclusions.

**Accuracy Vs Running Time:** It is rare to see any fair (same computational setting) empirical comparisons of SGD with existing adaptive SGD schemes, which compare the improvement in accuracy with respect to running time on the same computational platform. Almost all methods compare accuracy with the number of epochs, which is unfair to SGD which can complete $O(N)$ epochs at the computational cost (or running time) of one epoch for adaptive sampling schemes.

## 2 BACKGROUND

We first describe a recent advancement in the theory of sampling and estimation using locality sensitive hashing (LSH) (Indyk & Motwani, 1998) which will be heavily used in our proposal. Before we get into details of sampling, let us revise the two-decade-old theory of LSH.

### 2.1 LOCALITY SENSITIVE HASHING (LSH)

Locality-Sensitive Hashing (LSH) (Indyk & Motwani, 1998) is a popular, sub-linear time algorithm for approximate nearest-neighbor search. The high-level idea is to place similar items into the same bucket of a hash table with high probability. An LSH hash function maps an input data vector to an integer key

$$h(x) : \mathbb{R}^D \mapsto [0, 1, 2, \dots, N].$$

A collision occurs when the hash values for two data vectors are equal: $h(x) = h(y)$. The collision probability of most LSH hash functions is generally a monotonic function of the similarity

$$Pr[h(x) = h(y)] = \mathcal{M}(sim(x, y)),$$

where $\mathcal{M}$ is a monotonically increasing function. Essentially, similar items are more likely to collide with each other under the same hash fingerprint.

The algorithm uses two parameters, $(K, L)$. We construct $L$ independent hash tables from the collection $\mathcal{C}$. Each hash table has a meta-hash function $H$ that is formed by concatenating $K$ random independent hash functions from $\mathcal{F}$. Given a query, we collect one bucket from each hash table and return the union of $L$ buckets. Intuitively, the meta-hash function makes the buckets sparse and reduces the number of false positives, because only valid nearest-neighbor items are likely to match all $K$ hash values for a given query. The union of the $L$ buckets decreases the number of false negatives by increasing the number of potential buckets that could hold valid nearest-neighbor items.

The candidate generation algorithm works in two phases [See (Spring & Shrivastava, 2017) for details]:

1. **Pre-processing Phase:** We construct $L$ hash tables from the data by storing all elements $x \in \mathcal{C}$. We only store pointers to the vector in the hash tables because storing whole data vectors is very memory inefficient.

2. **Query Phase:** Given a query $Q$; we will search for its nearest-neighbors. We report the union from all of the buckets collected from the $L$ hash tables. Note, we do not scan all the elements in $\mathcal{C}$, we only probe $L$ different buckets, one bucket for each hash table.

After generating the set of potential candidates, the nearest-neighbor is computed by comparing the distance between each item in the candidate set and the query.

## 2.2 LSH for Estimations and Sampling

An item returned as candidate from a $(K, L)$-parameterized LSH algorithm (section 3.2) is sampled with probability $1 - (1 - p^K)^L$, where $p$ is the collision probability of LSH function. The LSH family defines the precise form of $p$ used to build the hash tables. This sampling view of LSH was first utilized to perform adaptive sparsification of deep networks in near-constant time, leading to efficient backpropagation algorithm (Spring & Shrivastava, 2016).

A year later, (Spring & Shrivastava, 2017) demonstrated the first theory of using these samples for unbiased estimation of partition functions in log-linear models. More specifically, the authors showed that since we know the precise probability of sampled elements $1 - (1 - p^K)^L$, we could design provably unbiased estimators using importance sampling type idea. This was the first demonstration that random sampling could be beaten with roughly the same computational cost as vanilla sampling. (Luo & Shrivastava, 2017) used the same approach for unbiased estimation of anomaly scoring function. (Charikar & Siminelakis) rigorously formalized these notions and showed provable improvements in sample complexity of kernel density estimation problems. Recently, (Chen et al., 2017) used the sampling in a very different context of connected component estimation for unique entity counts.

### 2.2.1 MIPS Sampling

Recent advances in maximum inner product search (MIPS) using asymmetric locality sensitive hashing has made it possible to sample large inner products.

For this paper, it is safe to assume that given a collection $\mathcal{C}$ of vectors and query vector $Q$, using $(K, L)$-parameterized LSH algorithm with MIPS hashing (Shrivastava & Li, 2014), we get a candidate set $S$ that every element $x_i \in \mathcal{C}$ is sampled with probability $p_i \leq 1$, where $p_i$ is a monotonically increasing function of $Q \cdot x_i$. Thus, we can pay a one-time linear cost of preprocessing $\mathcal{C}$ into hash tables, and any further adaptive sampling for query $Q$ only requires few hash lookups. We can also compute the probability of getting $x$.

Before getting into our main algorithm where we use the above sampling process for estimation, we would like to cover some of its properties. To begin with, the sampling scheme is not a valid distribution, i.e., $\sum_{x_i \in \mathcal{C}} p_i \neq 1$. In addition, given a query, the probability of sampling $x_i$ is not independent of the probability of sampling $x_j$ $(i \neq j)$. However, we can still use it for unbiased estimation. Details of such sampling are included in (Spring & Shrivastava, 2017). In fact, the form of sampling probability $p_i$ is quite unusual. $p_i$ is a monotonic function of $q \cdot x_i$ because $p_i = (1 - (1 - g(q \cdot x_i))^K)^L$, where $g(q \cdot x_i)$ is the collision probability.

## 3 The LSD Algorithm

### 3.1 A Generic Framework for Efficient Gradient Estimation

Our algorithm leverages the efficient estimations using locality sensitive hashing, which usually beats random sampling estimators while keeping the sampling cost near-constant. We first provide the intuition of our proposal, and the analysis will follow. Consider least squares regression with loss function $\frac{1}{N} \sum_{i=1}^{N} (y_i - \theta_t \cdot x_i)^2$, where $\theta_t$ is the parameter in the $t^{th}$ iteration. The gradient is just like a partition function. If we simply follow the procedures in (Spring & Shrivastava, 2017), we can easily show a generic unbiased estimator via adaptive sampling. However, better sampling alternatives are possible.

Observing that the gradient, with respect to $\theta_t$ concerning $x_i$, is given by $2(y_i - \theta_t \cdot x_i)x_i$, the $L_2$ norm of the gradient can therefore be written as an absolute value of inner product. according

to (Alain et al., 2015), the $L_2$ norm of the gradient is also the optimal sampling weight $w_i^*$ for $x_i$.

$$\|\nabla f(x_i, \theta_t)\|_2 = \left|2(\theta_t \cdot x_i - y_i)\|x_i\|_2\right| \tag{4}$$

$$= 2\left|\langle\theta_t, -1\rangle \cdot \langle x_i\|x_i\|_2, y_i\|x_i\|_2\rangle\right|, \tag{5}$$

where $\langle\theta_t, -1\rangle$ is a vector concatenation of $\theta$ with $-1$. If the data is normalized then we should sample $x_i$ in proportion to $w_i* = \left|\langle\theta_t, -1\rangle \cdot \langle x_i, y_i\rangle\right|$, i.e. large magnitude inner products should be sampled with higher probability.

As argued, such sampling process is expensive because $w_i^*$ changes with $\theta_t$. We address this issue by designing a sampling process that does not exactly sample with probability $w_i^*$ but instead samples from a different weighted distribution which is a monotonic function of $w_i^*$. Specifically, we sample from $w_i^{lsh} = f(w_i^*)$, where $f$ is some monotonic function. Before we describe the efficient sampling process, we first argue that a monotonic sampling is a good choice for gradient estimation.

---

**Algorithm 1** LSH-Sampled Stochastic gradient Descent (LSD) Algorithm

    **Input:** $D = x_i,\ y_i,\ N, \theta_0, \eta$
    **Input: LSH Family** $H$**, parameters** $K$**,** $L$
    **Output:** $\theta^*$
    $HT =$ Get preprocessed training data vectors $x_{lsh}, y_{lsh}$ and then put $\langle x_{lsh}^i,\ y_{lsh}^i\rangle$ into LSH Data structure.
    Get $x'_{train}, y'_{train}$ from preprocessed data
    $t = 0$
    **while** $NotConverged$ **do**
        $x_{lsh}^i,\ p = Sample(H, \text{HT}, K, \langle\theta_t, -1\rangle)$ (Algorithm 2)
        Get corresponding $x_{train}^{i'}, y_{train}^{i'}$ from preprocessed data
        $\theta_{t+1} := \theta_t - \eta_t\left(\frac{\nabla f(x_{train}^{i'}, \theta_t)}{p \times N}\right)$
    **end while**
    return $\theta^*$

---

For any monotonic function $f$, the weighted distribution $w_i^{lsh} = f(w_i^*)$ is still adaptive and changes with $\theta_t$. Also, due to monotonicity, if the optimal sampling prefers $x_i$ over $x_j$ i.e. $w_i^* \geq w_j^*$, then monotonic sampling will also have same preference, i.e., $w_i^{lsh} \geq w_j^{lsh}$.

The key insight is that there are two quantities in the inner product (equation 4), $\langle\theta_t, -1\rangle$ and $\langle x_i, y_i\rangle$. With successive iteration, $\langle\theta_t, -1\rangle$ changes while $\langle x_i, y_i\rangle$ is fixed. Thus, it is possible to preprocess $\langle x_i, y_i\rangle$ into hash tables (one time cost) and query with $\langle\theta_t, -1\rangle$ for efficient and adaptive sampling. With every iteration, only the query changes to $\langle\theta_{t+1}, -1\rangle$, but the hash tables remains the same. Few hash lookups are sufficient to sample $x_i$ for gradient estimation adaptively. Therefore, we only pay one-time preprocessing cost of building hash tables and few hash lookups, typically just one, in every iteration to get a sample for estimation.

There are few more technical subtleties due to the absolute value of inner product $\left|\langle\theta_t, -1\rangle \cdot \langle x_i, y_i\rangle\right|$, rather than the inner product itself. However, the square of the absolute value of the inner product

$$\left|\langle\theta_t, -1\rangle \cdot \langle x_i, y_i\rangle\right|^2 = T(\langle\theta_t, -1\rangle) \cdot T(\langle x_i, y_i\rangle),$$

can also be written as an inner product as it is a quadratic kernel, and $T$ is the corresponding feature expansion transformation. Again square is monotonic function, and therefore, our sampling is still monotonic as composition of monotonic functions is monotonic. Thus, technically we hash $T(\langle x_i, y_i\rangle)$ to create hash tables and the query at $t^{th}$ step is $T(\langle\theta_t, -1\rangle)$.

Once an $x_i$ is sampled via LSH sampling (Algorithm 2), we can precisely compute the probability of its sampling, i.e., $p_i$ (See section 2). It is not difficult to show that our estimation of full gradient is unbiased (Section 3.3).

### 3.2 ALGORITHMIC DETAILS

We first describe the detailed step of our gradient estimator in Algorithm 1. We also provide the sampling algorithm 2 with detail. Assume that we have access to the right LSH function $h$, and

---

**Algorithm 2** Sample

---

    **Input:** $H$ **(Hash functions)**, $HT[][]$ ($L$ **Hash Tables), K,** $Query$
    $cp(x, Q)$ **is the collision probability Pr(h(x)= h(Q)), under given LSH (known)**
    **Output: sampled data** $x$**, probability of sampling** $p$
    $l$, $S = 0$
    **while** true **do**
        $ti = random(1, L)$
        $bucket = H(Query, ti)$ (table specific hash)
        **if** HT[ti][bucket] = empty **then**
            $l$++
            continue;
        **end if**
        $S = |HT[ti][bucket]|$ (size of bucket)
        $x$ = randomly pick one element from $HT[ti][bucket]$
        break;
    **end while**
    $p = (1 - (1 - cp(x, Query)^K)^l) \times \frac{1}{S}$
    return $x$, $p$

---

its collision probability expression $cp(x, y) = Pr(h(x) = h(y))$. For linear regression, we can use signed random projections, simhash (Charikar, 2002), or MIPS hashing. With normalized data, simhash collision probability is $cp(x, y) = 1 - \frac{cos^{-1}(\frac{x \cdot y}{||x||_2 ||y||_2})}{\pi}$, which is monotonic in the inner product. Furthermore, we centered the data we need to store in the LSH hash table to make the simhash query more efficient.

### 3.2.1 RUNNING TIME OF SAMPLING

The computational cost of SGD sampling is merely a single random number generator. The cost of gradient update (equation 2) is one inner product, which is $d$ multiplications. If we want to design an adaptive sampling procedure that beats SGD, the sampling cost cannot be significantly larger than $d$ multiplications.

The cost of LSD sampling (Algorithm 2) is $K \times l$ hash computations followed by $l + 1$ random number generator, (1 extra for sampling from the bucket). Since the scheme works for any $K$, we can always choose $K$ small enough so that empty buckets are rare (see (Spring & Shrivastava, 2017)). In all of our experiments, $K = 5$ for which $l$ is almost always 1. Thus, we require $K$ hash computations and only two random number generations. If we use very sparse random projections, then $K$ hash computations only require a constant $\ll d$ multiplications. For example, in all our experiments we only need $\frac{d}{30}$ multiplication, in expectation, to get all the hashes using sparse projections. Therefore, our sampling cost is significantly less than $d$ multiplication which is the cost of gradient update. Using fast hash computation is critical for our method to work in practice.

### 3.2.2 NEAR-NEIGHBOR IS COSTLIER THAN LSH-SAMPLING

It might be tempting to use approximate near-neighbor search with query $\theta_t$ to find $x_i$. Near-neighbor search has been used in past Dhillon et al. (2011) to speed up coordinate descent. However, near-neighbor queries are expensive due to candidate generation and filtering. It is still sub-linear in $N$ (and not constant). Thus, even if we see epoch wise faster convergence, iterations with a near-neighbor query would be orders of magnitude slower than a single SGD iteration. Moreover, the sampling probability of $x$ cannot be calculated for near-neighbor search which would cause bias in the gradient estimates.

It is important to note that although LSH is heavily used for near-neighbor search, in our case, we use it as a sampler. For efficient near neighbor search, $K$ and $L$ grow with $N$ (Indyk & Motwani, 1998). In contrast, the sampling works for any $K$ and $l$ [1] as small as one leading to only approximately 1.5 times the cost of SGD iteration (see section 4). Efficient unbiased estimation is the key difference

---

[1]L represents the number of hash tables but l represents the number of hash tables used in one query

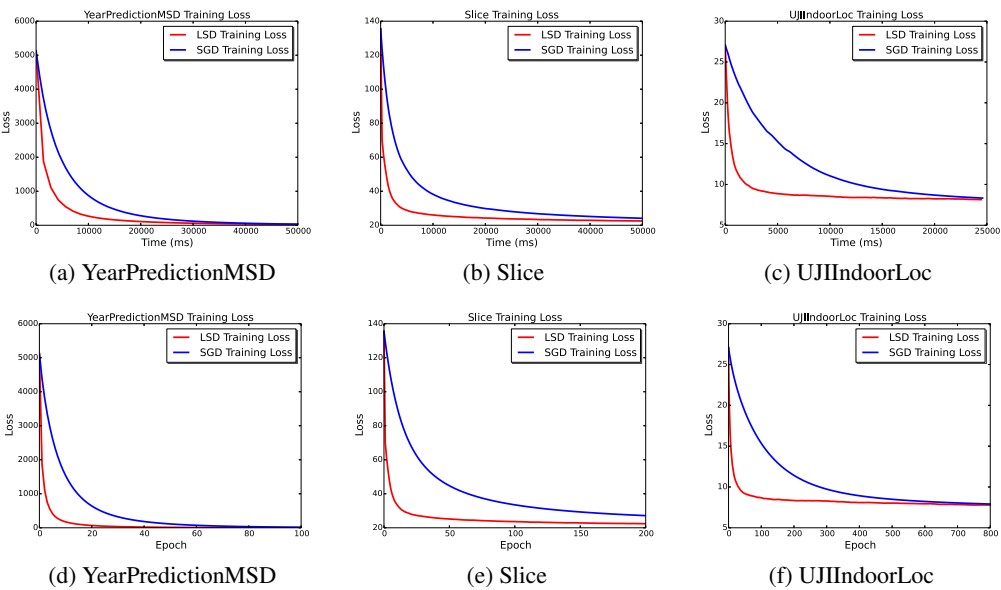

(a) YearPredictionMSD          (b) Slice          (c) UJIIndoorLoc

(d) YearPredictionMSD          (e) Slice          (f) UJIIndoorLoc

Figure 1: In subplots (a)(b)(c), the comparisons of Wall clock training loss convergence are made between plain LSD (red lines) and plain SGD (blue lines) separately in three datasets. We can clearly see the big gap between them representing LSD converge faster than SGD even in time wise. Subplots (c)(d)(e) shows the results for same comparisons but in epoch wise. We can see that LSD converges even faster than SGD which is not surprising because LSD costs a bit more time than SGD does in every iteration. Comparisons of testing loss are in supplemental material.

that makes sampling practical while near-neighbor query prohibitive. It is unlikely that a near-neighbor query would beat SGD in time, while sampling would.

## 3.3 VARIANCE ANALYSIS

In this section, we first prove that our estimator of the gradient is unbiased with lower variance than SGD for most real datasets. Define $S$ as the bucket that contains the sample $x$ from in Algorithm 2. For simplicity we denote the query as $\theta_t$ and $p_i = 1 - (1 - cp(x_i, \theta_t)^K)^l$ as the probability of finding $x_i$ in bucket $S$.

**Theorem 1.** *The following expression is an unbiased estimator of the full gradient*

$$Est = \frac{1}{N} \sum_{i=1}^{N} \mathbb{1}_{x_i \in S} \mathbb{1}_{(x_i = x_m | x_i \in S)} \frac{\nabla f(x_i, \theta_t) \cdot |S|}{p_i}, \tag{6}$$

$$\mathbb{E}[Est] = \frac{1}{N} \sum_{i=1}^{N} \nabla f(x_i, \theta_t). \tag{7}$$

**Theorem 2.** *The Trace of the covariance of our estimator:*

$$Tr(\Sigma(Est)) = \frac{1}{N^2} \sum_{i=1}^{N} \frac{\|\nabla f(x_i, \theta_t)\|_2^2 \cdot |S|}{p_i} - \frac{1}{N^2} \|(\sum_{i=1}^{N} \nabla f(x_i, \theta_t))\|_2^2 \tag{8}$$

The trace of the covariance of LSD is the total variance of the descent direction. The variance can be minimized when the sampling probability of $x_i$ is proportional to the $L_2$-norm of the gradient we mentioned in Section 1.1. The intuition of the advantage of LSD estimator comes from sampling $x_i$ under a distribution monotonic to the optimal one. We first make a simple comparison of the variance of LSD with that of SGD theoretically and then in Section 4 and we would further empirically show the drastic superiority of LSD over SGD.

**Lemma 1.** *The Trace of the covariance of LSD's estimator is smaller than that of SGD's estimator if*

$$\frac{1}{N} \sum_{i=1}^{N} \frac{\|\nabla f(x_i, \theta_t)\|_2^2 \cdot |S|}{p_i} < \sum_{i=1}^{N} \|\nabla f(x_i, \theta_t)\|_2^2, \tag{9}$$

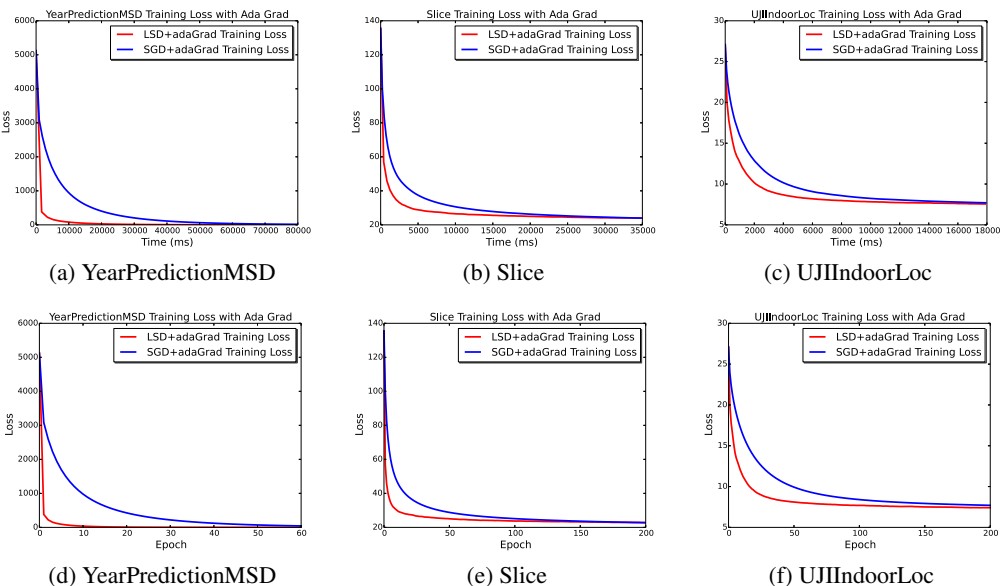

(a) YearPredictionMSD      (b) Slice      (c) UJIIndoorLoc

(d) YearPredictionMSD      (e) Slice      (f) UJIIndoorLoc

Figure 2: In subplots (a)(b)(c), the comparisons of Wall clock training loss convergence are made between LSD+adaGrad (red lines) and SGD+adaGrad (blue lines) separately in three datasets. We can again see the similar gap between them representing LSD converge faster than SGD in time wise. Subplots (c)(d)(e) show the results for same comparisons but in epoch wise. We can see that LSD converges even faster than SGD. Comparisons of testing loss are in supplemental material.

SGD would perform well if the data is uniformly distributed but it is unlikely in practice. Recall that the collision probability $p_i = 1 - (1 - p^K)^l$ mentioned in Section 2.2. Note that $l$ here according to Algorithm 2 is the number of tables that have been utlized by the sampling process. In most practical cases and also in our experiment, $K$ and $l$ are relatively small. L should be large to ensure enough randomness but it does not show up in the sampling time (See Alg. 2). LSD can be efficient and achieve a much smaller variance than SGD by setting small values of $K$ and $l$. It is not difficult to see that if several terms in the summation satisfy $\frac{|S|}{p_i N} \leq 1$, then the variance of our estimator is better than random sampling. If the data is clustered nicely, i.e. a random pair has low similarity, by tuning $K$, we can achieve the above inequality of $|S|$, $p_i$ and N. See Spring & Shrivastava (2017); Charikar & Siminelakis for more details on when LSH sampling is better than random sampling.

## 4 EXPERIMENTS

We examine the effectiveness of our algorithm on three large regression dataset, in the area of musical chronometry, clinical computed tomography, and WiFi-signal localization, respectively. The dataset descriptions and our experiment results are as follows:

**YearPredictionMSD:** (Lichman, 2013) The dataset contains 515,345 instances subset of the Million Song Dataset with dimension 90. We respect the original train/test split, first 463,715 examples for training and the remaining 51,630 examples for testing, to avoid the 'producer effect' by making sure no song from a given artist ends up in both the train and test set.

**Slice:** (Lichman, 2013) The data was retrieved from a set of 53,500 CT images from 74 different patients. It contains 385 features. We use 42,800 instances as training set and the rest 10,700 instances as the testing set.

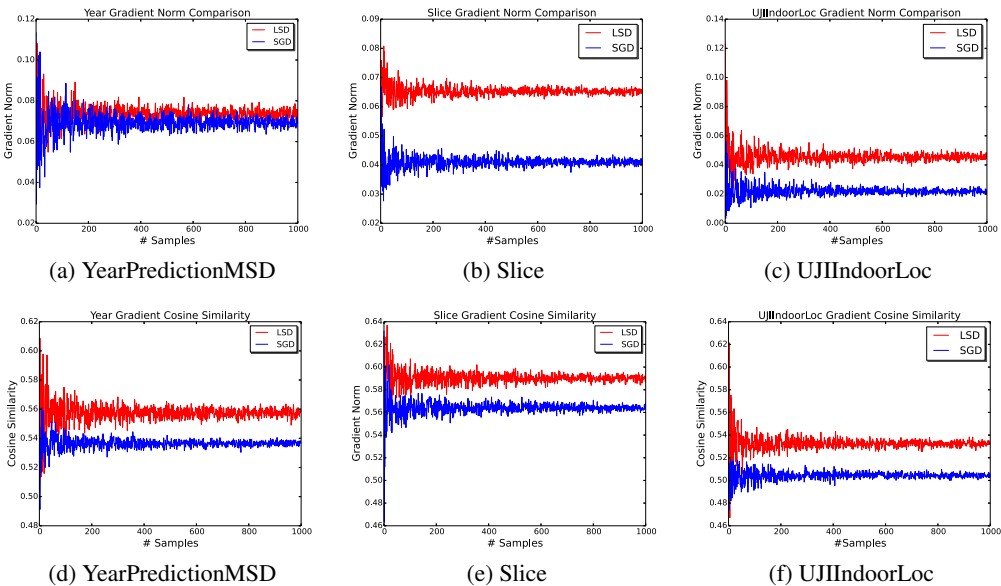

Figure 3: Norm and cosine similarity comparisons of LSD and SGD gradient estimation. Subplots (a)(b)(c) show the comparisons of the average (over number of samples) gradient $L_2$ norm of the points that LSD (red lines) and SGD sampled (blue lines). As argued before, LSD samples with probability monotonic to $L_2$ norm of the gradients while SGD samples uniformly. It matches with the results shown in the plots that LSD queries points with larger gradient than SGD does. Subplots (d)(e)(f) show the comparison of the cosine similarity between gradient estimated by LSD and the true gradient and the cosine similarity between gradient estimated by SGD and the true gradient. Note that the variance of both norm and cosine similarity reduce when we average over more samples.

**UJIIndoorLoc:** (Torres-Sospedra et al., 2014) The database covers three buildings of Universitat Jaume I with 4 or more floors and almost 110,000 $m^2$. It is a collection of 21,048 indoor location information with 529 attributes containing the WiFi fingerprint, the coordinates where it was taken, and other useful information. We equally split the total instances for training and testing.

All datasets were preprocessed as described in Section 3.2. Note that for all the experiments, the choice of the gradient decent algorithm was the same. For both SGD and LSD, the only difference in the gradient algorithm was the gradient estimator. For SGD, a random sampling estimator was used, while for LSD, the estimator used the adaptive estimator. We used fixed values $K = 5$ and $L = 100$ for all the datasets. $l$ is the number of hash tables that have been searched before landing in a non-empty bucket in a query. In our experiments $l$ is almost always as low as 1. $L$ only affects preprocessing but not sampling. Our hash function was simhash (or signed random projections) and we used sparse random projections with sparsity $\frac{1}{30}$ for speed. We know that epoch wise convergence is not a true indicator of speed as it hides per epoch computation. Our main focus is convergence with running time, which is a better indicator of computational efficiency.

To the best of our knowledge, there is no other adaptive estimation baseline, where the cost of sampling per iteration is less than linear $O(N)$. Since our primary focus would be on wall clock speedup, no $O(N)$ estimation method would be able to outperform $O(1)$ SGD (and LSD) estimates on the same platform. From section 3.2.2, even methods requiring a near-neighbor query would be too costly (orders of magnitude) to outperform SGD from computational perspective.

## 4.1 LSD vs. SGD

In the first experiment, we compare vanilla SGD with LSD, i.e., we use simple SGD with fixed learning rate. This basic experiment aims to demonstrate the performance of pure LSD and SGD

without involving other factors like $L_1/L_2$ regularizations on linear regression task. In such a way, we can quantify the superiority of LSD.

Figure 1 shows the decrease in the squared loss error with epochs. Blue lines represent SGD and red lines represent LSD. It is obvious that LSD converges much faster than SGD in both training and testing loss comparisons. This is not surprising with the claims in Section 3.2.1 and theoretical proof in Section 3.3. Since LSD uses slightly more computations per epoch than SGD does, it is hard to defend if LSD gains enough benefits simply from the epoch wise comparisons. We therefore also show the decrease in error with wall clock time also in figure 1. Wall clock time is the actual quantification of speedups. Again, on every single dataset, LSD shows faster time-wise convergence as well. (Plots for Testing loss are in the supplementary material.)

### 4.2 LSD+ADAGRAD VS. SGD+ADAGRAD

As argued in section 1.1, our LSD algorithm is complimentary to any gradient-based optimization algorithm. We repeated the first experiment but using AdaGrad (Duchi et al., 2011) instead of plain SGD. Again, other settings are fixed for both algorithms but the only change in the competing algorithm is the gradient estimates per epoch. Figure 2 shows epoch wise and running time comparisons on LSD and SGD convergence. The trends as expected are similar to those of LSD vs. SGD. LSD with AdaGrad outperforms AdaGrad (SGD) estimates of gradients both epoch-wise and time-wise. (Plots for Testing loss are in the supplementary.)

### 4.3 LSD, SGD VS. TRUE GRADIENT:

In this section, as a sanity check, we first verify weather LSD samples data point with probability monotonic to $L_2$ norm of the gradient mentioned in section 3.1. In order to do that, we freeze the optimization at an intermediate iteration and use the $\theta$ at that moment to sample data points with LSD as well as SGD to compute gradient $L_2$ norm separately. The upper three plots in Figure 3 show the comparison of the sampled gradient norm of LSD and SGD. X-axis represents the number of samples that we averaged in the above process. It is obvious that LSD sampled points have larger gradient norm than SGD ones consistently across all three datasets.

In addition, we also do a sanity check that if empirically, the chosen sample from LSD get better estimation of the true gradient direction than that of SGD. Again, we freeze the program at an intermediate iteration like the experiments above. Then we compute the angular similarity of full gradient (average over the training data) direction with both LSD and SGD gradient direction, where, $Similarity = 1 - \frac{cos^{-1}\frac{x \cdot y}{\|x\|_2 \|y\|_2}}{\pi}$. From the bottom three plots in Figure 3, we can see that in average, LSD estimated gradient has smaller angle (more aligned) to true gradient than SGD estimated gradient.The variance of both norm and cosine similarity reduce when we average them over more samples as shown in plots.

## 5 CONCLUSION

In this paper, we proposed a novel LSH-based sampler with a reduction to the gradient estimation variance. We achieved it by sampling with probability proportional to the $L_2$ norm of the instances gradients leading to an optimal distribution that minimizes the variance of estimation. More remarkably, LSD is as computationally efficient as SGD but achieves faster convergence not only epoch wise but also time wise.

## 6 ACKNOWLEDGEMENTS

This work was supported by National Science Foundation IIS-1652131, RI-1718478, and a GPU grant from NVIDIA.

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

## A    EPOCH PLOTS AND PROOFS

**Theorem 3.** *Let $S$ be the bucket that sample $x_m$ is chosen from in Algorithm 2. Let $p_m$ be the sampling probability associated with sample $x_m$. Suppose we query a sample with $\theta_t$. Then we have an unbiased estimator of the full gradient:*

$$Est = \frac{1}{N} \sum_{i=1}^{N} \mathbb{1}_{x_i \in S} \mathbb{1}_{(x_i = x_m | x_i \in S)} \frac{\nabla f(x_i, \theta_t) \cdot |S|}{p_i}$$

$$\mathbb{E}[Est] = \frac{1}{N} \sum_{i=1}^{N} \nabla f(x_i, \theta_t)$$

*Proof.*

$$\mathbb{E}[\mathbb{1}_{x_i \in S}] = p_i, \quad and \quad \mathbb{E}[\mathbb{1}_{x_i = x_m | x_i \in S}] = \frac{1}{|S|}.$$

Also note that

$$\mathbb{E}[\mathbb{1}_{x_i \in S} \mathbb{1}_{x_i = x_m | x_i \in S}] = \mathbb{E}[\mathbb{1}_{x_i \in S}] \mathbb{E}[\mathbb{1}_{x_i = x_m | x_i \in S}].$$

Then,

$$\begin{aligned}
\mathbb{E}[Est] &= \frac{1}{N} \mathbb{E}\left[\sum_{i=1}^{N} \mathbb{1}_{x_i \in S} \mathbb{1}_{x_i = x_m | x_i \in S} \frac{\nabla f(x_i, \theta_t) \cdot |S|}{p_i}\right] \\
&= \frac{1}{N} \sum_{i=1}^{N} \mathbb{E}[\mathbb{1}_{x_i \in S} \mathbb{1}_{x_i = x_m | x_i \in S}] \cdot \mathbb{E}\left[\frac{\nabla f(x_i, \theta_t) \cdot |S|}{p_i}\right] \\
&= \frac{1}{N} \sum_{i=1}^{N} p_i \cdot \frac{1}{|S|} \frac{\nabla f(x_i, \theta_t) \cdot |S|}{p_i} \\
&= \frac{1}{N} \sum_{i=1}^{N} \nabla f(x_i, \theta_t)
\end{aligned}$$

$\square$

**Theorem 4.** *The Trace of the covariance of our estimator is:*

$$Tr(\Sigma(Est)) = \frac{1}{N^2} \sum_{i=1}^{N} \frac{\|\nabla f(x_i, \theta_t)\|_2^2 \cdot |S|}{p_i} - \frac{1}{N^2} \left(\sum_{i=1}^{N} \|\nabla f(x_i, \theta_t)\|_2\right)^2.$$

*Proof.*

$$Tr(\Sigma(Est) = \mathbb{E}[Est^T Est] - \mathbb{E}[Est]^T \mathbb{E}[Est]$$

$$Est^T Est = \frac{1}{N^2} \sum_{i,j}^{N} \mathbb{1}_{x_i \in S} \mathbb{1}_{x_j \in S} \mathbb{1}_{x_i = x_m | x_i \in S} \mathbb{1}_{x_j = x_m | x_j \in S} \frac{\nabla f(x_i, \theta_t) \cdot \nabla f(x_j, \theta_t) \cdot |S|^2}{p_i \cdot p_j}$$

$$= \frac{1}{N^2} \sum_{i}^{N} \mathbb{1}_{x_i \in S} \cdot \mathbb{1}_{x_i = x_m | x_i \in S} \frac{(\nabla \| f(x_i, \theta_t)))\|_2^2 \cdot |S|^2}{p_i^2}$$

$$\mathbb{E}[Est^T Est] = \frac{1}{N^2} \sum_{i}^{N} \frac{(\| \nabla f(x_i, \theta_t)))\|_2^2 \cdot |S|}{p_i}$$

$$Tr(\Sigma(Est)) = \frac{1}{N^2} \sum_{i=1}^{N} \frac{\| \nabla f(x_i, \theta_t) \|_2^2 \cdot |S|}{p_i} - \frac{1}{N^2} (\sum_{i=1}^{N} \nabla \| f(x_i, \theta_t) \|_2)^2$$

$\square$

**Lemma 2.** *The Trace of the covariance of LSD's estimator is smaller than that of SGD's estimator if*

$$\frac{1}{N} \sum_{i=1}^{N} \frac{\| \nabla f(x_i, \theta_t) \|_2^2 \cdot |S|}{p_i} < \sum_{i=1}^{N} \| \nabla f(x_i, \theta_t) \|_2^2 \tag{10}$$

*Proof.* The trace of covariance of regular SGD is

$$Tr(\Sigma(Est')) = \frac{1}{N} \sum_{i}^{N} \| \nabla f(x_i, \theta_t) \|_2^2 - \frac{1}{N^2} (\sum_{i=1}^{N} \| \nabla f(x_i, \theta_t) \|_2)^2. \tag{11}$$

By (11) and (17), one can easily see that $Tr(\Sigma(Est)) < Tr(\Sigma(Est'))$ when (16) satisfies. $\square$

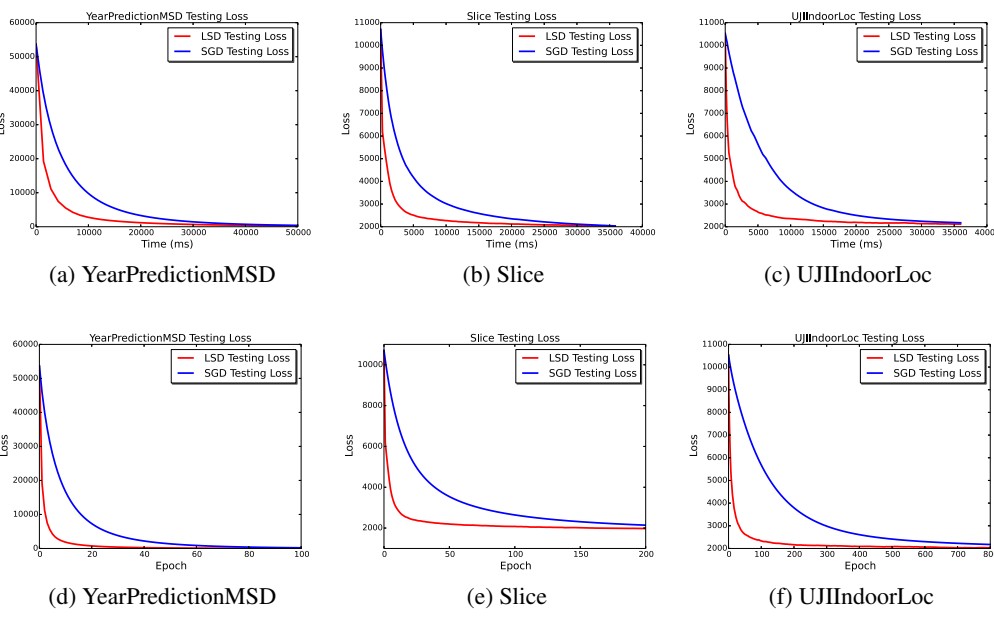

(a) YearPredictionMSD        (b) Slice        (c) UJIIndoorLoc

(d) YearPredictionMSD        (e) Slice        (f) UJIIndoorLoc

Figure 4: Epoch wise convergence comparisons of training and testing loss of LSD and SGD.

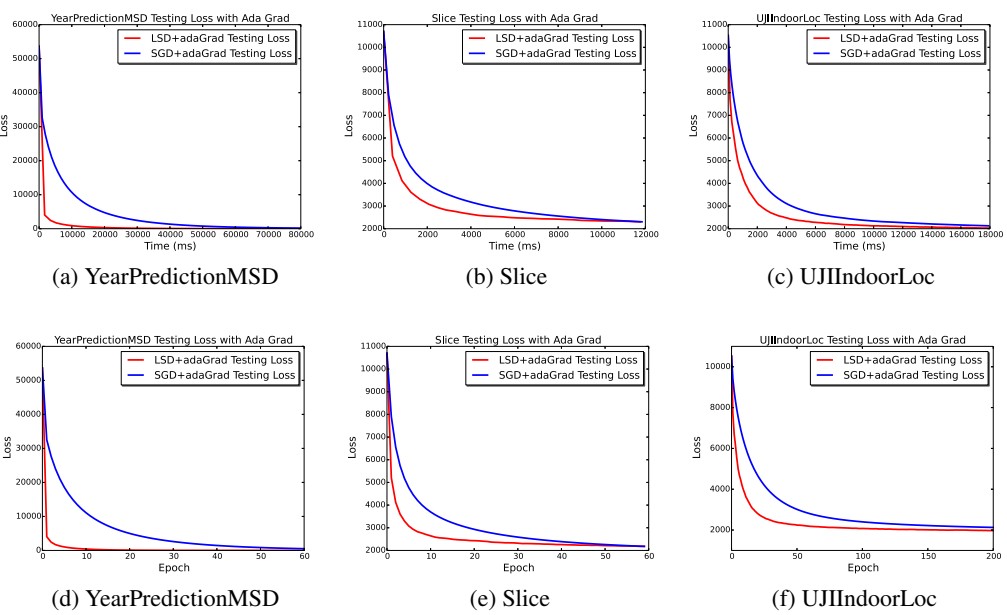

Figure 5: Epoch wise convergence comparisons of training and testing loss of LSD+adaGrad and SGD+adaGrad.

