# OpenReview forum: "LSH-SAMPLING BREAKS THE COMPUTATIONAL CHICKEN-AND-EGG LOOP IN ADAPTIVE STOCHASTIC GRADIENT ESTIMATION"
_ICLR.cc/2018/Conference — Invite to Workshop Track_

### Official Review · AnonReviewer2 · 2017-11-27
**Creative Paper Worth Sharing**

**Rating:** 8
**Confidence:** 4

**Review:**

Authors propose sampling stochastic gradients from a monotonic function proportional to gradient magnitudes by using LSH. I found the paper relatively creative and generally well-founded and well-argued.

Nice clear example with least squares linear regression, though a little hard to tell how generalizable the given ideas are to other loss functions/function classes, given the authors seem to be taking heavy advantage of the inner product.

Experiments: appreciated the wall clock timings.

SGD comparison: “fixed learning rate.” Didn't see how the initial (well constant here) step size was tuned? Why not use the more standard 1/t decay?

Fig 1: Suspicious CIFAR100 that test objective is so much better than train objective? Legend backwards?

Why were so many of the chosen datasets have so few training examples?

Paper is mostly very clearly written, though a bit too redundant and some sentences are oddly ungrammatical as if a word is missing - just needs a careful read-through.

---

> ### Author Response · Authors · 2017-12-23
> **thanks for encouraging comment**
>
> Thanks for the encouraging comment.
> We are happy to get your support and hope you will clarify the misconception of other reviewers in the subsequent discussions.
>
> To avoid any bells and whistles, we show plain SGD as well as adagrad which adaptively chooses the step size based on the previous gradients estimates.  We did not tune anything to nullify the effect of any tuning and ensure an apples-to-apples comparison.  Better gradient estimate leads to improvements despite SGD or adagrad.
>
> Inner product naturally goes for linear regression as well as logistic (exp^{inner product}).  A natural next step is to look at popular loss function as well as existing LSH to see if there are other sweet spots.
>
> Other than CIFAR, we chose high dimensional regression datasets  (not classification) from UCI. https://archive.ics.uci.edu/ml/datasets.html unfortunately, all high dimensional regressions datasets are small. Let us know if you have any suggestions on that.

---

### Official Review · AnonReviewer1 · 2017-11-28
**Another application of LSH sampling**

**Rating:** 4
**Confidence:** 5

**Review:**

  The main idea in the paper is fairly simple:

 The paper considers SGD over an objective of the form of a sum over examples of a quadratic loss.
The basic form of SGD selects an example uniformly.   Instead,  one can use any probability distribution over examples and apply inverse probability weighting to retain unbiasedness of the gradient.

  A good method (that builds on classic pps sampling) is to select examples with higher normed gradients with higher probability [Alain et al 2015].

  With quadratic loss,  the gradient increases with the inner product of the parameter vector (concatenated with -1) and the example vector x_i (concatenated with the label y_i).

  For the current parameter vector \theta,  we would like to sample examples so that the probability of sampling larger inner products is larger.

  The paper uses LSH structures, computed over the set of examples,
 to quickly sample examples with large inner products with the current parameter vector \theta.   Essentially, two vectors are hashed to the same bucket with probability that increases with their cosine similarity.
 So we select examples in the same LSH bucket as \theta (for rubstness, we use multiple LSH mappings).


strengths:  simple idea that can work well in the context of sampling examples for SGD

weaknesses:

  The novelty in the paper is limited. The use of LSH for sampling is a common technique to sample more similar vectors with higher probability.  There are theorems,  but they are trivial, straightforward applications of importance sampling.

 The paper is not well written. The presentation is much more complex that need be. References to classic weighted sampling are

  The application is limited to certain loss functions for which we can compute LSH structures.  This excludes NN models and even the addition of regularization to the quadratic loss can affect the effectiveness.

---

> ### Author Response · Authors · 2017-12-23
> **This is first time LSH is used for unbiased gradient Estimations (with near-constant amortized adaptive sampling cost like SGD)**
>
>   The novelty in the paper is limited. The use of LSH for sampling is a common technique to sample more similar vectors with higher probability.  There are theorems,  but they are trivial, straightforward applications of importance sampling.
> >>   LSH as sampling was first used very recently (early 2016).
> Note the importance weighting factor in the algorithm of 1 - (1-p^K)^L. It is about the unbiased estimation of gradients rather than a simple heuristic.
>
> We challenge the reviewer to show one paper which shows the use of LSH as sampling for unbiased estimation of the gradient in SGD.
>
> Simplicity is not bad, especially when it beats a fundamental barrier.
>
> **************
>
> The paper uses LSH structures, computed over the set of examples,
>  to quickly sample examples with large inner products with the current parameter vector \theta.   Essentially, two vectors are hashed to the same bucket with probability that increases with their cosine similarity.
>  So we select examples in the same LSH bucket as \theta (for robustness, we use multiple LSH mappings).
> >> Not really, the process is about unbiased estimation (mentioned in the paper at several places). Again you are missing the importance style weights. And the sampling is correlated and not normalized, so it is something never seen before. Due to the simplicity of our proposal, it might be easy to overlook the subtlety of the methods.
>
> We reiterate, this not yet another heuristic here. For the first time, we see some hope of beating SGD in running time using better estimator, and this does not happen often.
>
> We hope these comments will lead to a healthy discussion and correction of any misconceptions on either side :)
>
> Thanks for taking time in trying to improve our paper.

---

> > ### Comment · AnonReviewer1 · 2018-01-06
> > **response**
> >
> > >> Again you are missing the importance style weights. And the sampling is correlated and not normalized, so it is something never seen before
> >
> > I do not believe I missed the unbiasedness.   Note that "importance weights" (inverse probability weighting)  is a 7 decades old technique to obtain unbiased estimators from unequal probability samples.   When the probabilities are better "correlated" with the weights (similarity) the variance is better.
> >
> > The unnormalised sampling (based on weights without knowing the sum) is also decades old.  Say order sampling.
> >
> >  To put history in perspective.  LSH schemes are essentially sampling scheme.  There are many older techniques that simply perform similarity-based sampling and did not call it LSH.  The beautiful theory of LSH from the last two decades was about relating the sampling schemes to approximate NN structures.    What the very recent work does is using LSH sampling schemes, again, for sampling...    That recent thread is very nice as it notes this in the context of some new applications, with some very nice analysis.
> >
> > I believe that current submission novelty is really only in noting the potential SGD application.   A very convincing demonstration of the potential of that, with comparison to other methods, and proper presentation, could make a very nice paper.  I am looking forward to see the next version.

---

> > > ### Author Response · Authors · 2018-01-06
> > > **None of the historical sampling is efficient**
> > >
> > > I do not believe I missed the unbiasedness.   Note that "importance weights" (inverse probability weighting)  is a 7 decades old technique to obtain unbiased estimators from unequal probability samples.   When the probabilities are better "correlated" with the weights (similarity) the variance is better.
> > >
> > > The unnormalised sampling (based on weights without knowing the sum) is also decades old.  Say order sampling.
> > > >>  Yes, however, any non-trivial (interesting) sampling is O(N) as simply computing any weight requires O(N) cost per iteration.  LSH is the only way to get is constant amortized cost
> > >
> > > I believe that current submission novelty is really only in noting the potential SGD application.
> > > >> Isn't is a neat observation? We are really  excited about this striking possibility. What in the world gives constant time adaptive sampling? Any form of adaptiveness is O(N), except a wierd mathematical form of 1 - (1-p^K)^L (unheard of) which admits contant amortized cost sampling and at the same time is adaptive.
> > >
> > >  To put history in perspective.  LSH schemes are essentially sampling scheme.  There are many older techniques that simply perform similarity-based sampling and did not call it LSH.
> > > >> Computing similarities itself is O(N) to start as there are N data points.
> > >
> > >  The beautiful theory of LSH from the last two decades was about relating the sampling schemes to approximate NN structures.
> > > >> LSH as sampling just came in 2016 not last decade. Until that time LSH was thought to be a fast subroutine for  NN search and its potential as a sampler and unbiased estimator were not heard of.  The beauty is that sampling can be amortized constant time, which was first shown in early 2016. We are not aware of any literature that uses LSH as samplers before that.
> > >
> > >
> > >   A very convincing demonstration of the potential of that, with comparison to other methods, and proper presentation, could make a very nice paper.  I am looking forward to see the next version.
> > > >> Thanks for the encouragement. Is there anything you have in mind, and we will compare it. We know that beating SGD on running time (with same resources) is hard, so it looks rather easy for us.
> > >
> > > We hope you will support our paper.  We are happy to do any additional comparisons you have in mind.

---

> > > ### Author Response · Authors · 2018-01-10
> > > **One subtlety about sampling Vs near neighbor (Sampling is way more efficient, NNbor is unlikely to beat SGD in wall clock time)**
> > >
> > > I forgot to mention that near neighbor queries are significantly slower than sampling.
> > >
> > > In our experiments sampling requires only one memory lookup and random number generation
> > >
> > > On the contrary,  near-neighbor query (per update) require in theory to probe n^\rho (grows with data) lookup, followed by bucket aggregation, followed by filtering using distance computations (again of the order n^\rho).
> > >
> > > Although \rho < 1 (sublinear) but sill compared to SGD (one random sample) this process (or any near neighbor query) is unlikely  to lead to a faster in running time algorithm.
> > >
> > > This is the reason; any neighbor based sampling approach is unlikely to beat SGD in running time.  While ours can! (only one lookup, no costly candidate filtering)
> > >
> > > We hope you see the critical subtlety with this new view of LSH.

---

> > > > ### Comment · AnonReviewer1 · 2018-01-10
> > > > **NO  "new view"  of LSH....    but application is nice.  Build on your strengths but don't oversell**
> > > >
> > > > Ok, lets put things straight.
> > > >
> > > > LSH, that is, sampling schemes were more similar entities are more likely to be sampled together, are  known for decades.  E.g., based on random projections or on consistent samples.
> > > >
> > > >  The "big deal" about the theory of LSH (2 decades) was a very general method of using these "weak" sampling schemes to construct approximate NN structures.
> > > >
> > > >  What you are doing is using LSH sampling schemes for exactly what they are...  weighted sampling by similarity.
> > > >
> > > >   BTW,   I am pretty confident that you do not want a NN here (largest gradient norm) even if you could get it for free.    In particular, this can very badly bias the expectation of the gradient  and you will lose theoretical convergence properties.  My hunch is that it would also be very bad in practice.
> > > >
> > > >
> > > >  The novelty in your paper,  which could make for a  **very  nice** application,  is the simple observation that LSH can be applied to select examples that have larger gradients in the context of GD with quadratic loss (because then you have the LSH function (cosine similarity between the parameter of the model and the vectors).
> > > >
> > > >    I believe that if this is written well,  explaining what is  the contribution (this observation and experiments),  careful evaluation,  providing clarity to readers without much background, point on  the limitations, present the simple idea for what it is,  take credit only for what you contribute,  write it in a way that provide value to readers, then it can make a very nice paper.

---

> > > > > ### Author Response · Authors · 2018-01-10
> > > > > **Sampling as well as sub-linearity.  Just sampling is simple but expensive. With LSH and only using hash tables they can be made efficient.**
> > > > >
> > > > > LSH, that is, sampling schemes were more similar entities are more likely to be sampled together, are known for decades.  E.g., based on random projections or on consistent samples.
> > > > >
> > > > >  What you are doing is using LSH sampling schemes for exactly what they are...  weighted sampling by similarity.
> > > > > >> Point out any earlier literature exploiting LSh for sub-linear adaptive sampling given a query? Unbiased estimation with LSH in sublinear time is not known before.
> > > > >
> > > > > The new thing is sampling in sub-linear time that requires indexing, and simply random projections won't help.
> > > > >  Random projections are good for estimation (not sub-linear in the number of examples) unless combined with quantizations and indexing.  We can stress this part more if needed. It is easy to miss.  It requires data structures.
> > > > >
> > > > > Any non-trivial similarity based adaptive sampling (using random projection or otherwise) is a linear cost without indexing (hash tables). Its the power of data structure combined with properties of random projections.  The power of data structure is often missed with LSH and dimensionality reduction is thought to be the prime reason.
> > > > >
> > > > >
> > > > >
> > > > >    I believe that if this is written well,  explaining what is  the contribution (this observation and experiments),  careful evaluation,  providing clarity to readers without much background, point on  the limitations, present the simple idea for what it is,  take credit only for what you contribute,  write it in a way that provide value to readers, then it can make a very nice paper.
> > > > > >>  We are happy to make any suggested changes, as we can clearly see that LSH is so widely popular that the important points can be easily lost.
> > > > > We hope you see we are not claiming for more than what we are contributing.

---

> > > > > > ### Comment · AnonReviewer1 · 2018-01-11
> > > > > > **response**
> > > > > >
> > > > > > Response to the following comments/questions:
> > > > > >
> > > > > > > >The new thing is sampling in sub-linear time that requires indexing, and simply random projections won't help.
> > > > > > >>Any non-trivial similarity based adaptive sampling (using random projection or otherwise) is a linear cost without indexing (hash tables). Its the power of data structure combined with properties of random projections.  The power of data structure is often missed with LSH and dimensionality reduction is thought to be the prime reason.
> > > > > > >> Point out any earlier literature exploiting LSh for sub-linear adaptive sampling given a query?
> > > > > >
> > > > > > 1.  This "indexing" is simply bucketing by hash/randomized function.   AND it is not new! It is the first very basic step of turning the randomized similarity functions into approx NN schemes.  This component of LSH was also used for similarity applications for decades.  For example, it is routinely used to generate graphs from massive metric data (kNN is too expensive).  In this context think of a node as a "query" if you wish and you connect it to sampled nodes from LSH buckets.   This creates a graph where closer points are much more likely to have an edge connected.
> > > > > >
> > > > > > 2. In any case,  this is not even claimed to be a contribution of this submission.  Only that the presentation of this paper seems to attribute basic known methods to very recent work.
> > > > > >
> > > > > >
> > > > > > This is not the only issue to correct.  You can make a nice paper,  but the submission is not yet there.

---

> > > > > > > ### Author Response · Authors · 2018-01-13
> > > > > > > **Thanks for encouraging comments**
> > > > > > >
> > > > > > > Thanks for the discussions!
> > > > > > >
> > > > > > > We will restress the subtleties and differenced of indexing, sub-linear similarity search, and the new line of sub-linear adaptive sampling and unbiased estimation in any future versions of the paper.
> > > > > > >
> > > > > > > Let us know if you think anything else will be helpful.

---

### Official Review · AnonReviewer3 · 2017-12-04
**A simple application of LSH but logically disordered**

**Rating:** 4
**Confidence:** 5

**Review:**

The main contribution of this work is just a combination of LSH schemes and SGD updates. Since hashing schemes essentially reduce the dimension, LSH brings computational benefits to the SGD operation. The targeted issue is fundamentally important, and the proposed approach (exploiting LSH schemes) seems to be sound.  Specifically, LSH schemes fit into the SGD schemes since they hash two vectors to the same bucket with probability in proportional to their distance (here, inner product or Cosine similarity).

Strengths:  a sound approach; a simple and straightforward idea that is shown to work well in evaluations.

Weaknesses:
1. The phrase of "computational chicken-and-egg loop" in the title and also in the main body is misleading and not accurate. The so-called "chicken-and-egg” issue concerns the causality dilemma: two causally related things, which comes the first. In the paper, the authors concerned "more accurate gradients" and "faster convergence"; their causality is very clear (the first leads to the second), and there is no causality dilemma. Even from a computational perspective, "SDG schemes aim for computational efficiency" and "stochastic makes the convergence slow down" are not a causality dilemma.  The reason behind is that the latter is the cost of the first one, just the old saying that "there is no such thing as a free lunch". Therefore, this disordered logic makes the title very misleading, and all the corresponding descriptions in the main body are obscured by "twisted" and unnatural logics.

2. The depth is so limited. Besides a good observation that LSH fits well into SDG, there are no more in-depth results provided. The theorems (Theorems 1~3) are trivial, with loose relations with LSH.

3. The LSH schemes are not correctly referred to. Since the similarity metric is inner-product, the authors are expected to refer to Cosine similarity and inner-product based LSHs, which were published recently in NIPS. It is not in depth to assume "any known LSH scheme" in Alg. 2. Accordingly again, Theorems 1~3 are unrelated with this specific kind of similarity metric (Cosine similarity).

4. As the authors tried hard to stick to the unnecessary (a bit bragging) phrase "computational chicken-and-egg loop", the organization and presentation of the whole manuscript are poor.

5. Occasionally, there are typos, and it is not good to use words in formulas. Please proof-read carefully.

---

> ### Author Response · Authors · 2017-12-23
> **We disagree with most but can change wordings if that is the main concern,**
>
> Since hashing schemes essentially reduce the dimension, LSH brings computational benefits to the SGD operation
> >> NO .... Not at all.  It has nothing to do with dimensionality reduction at all. It is about efficient sampling using hash tables. (Also see response to AnonReviewer1)
>
> We are afraid that the reviewer is mistaken as to what the method it, despite this being mentioned at several placed very explicitly. We still try our best to respond to concerns.
>
> 1)   SGD reduces the costly iteration (O(1) per iteration) but increases the number of iterations. Any known adaptive scheme to reduce the number of iterations leads to very costly O(N) per iteration. We refer this inherent tradeoff as chicken and egg loop. If this is a big issue, we can easily change it?
>
> 2) See response to AnonReviewer1. Missing the subtlety of the algorithm is easy. Simplicity that beats a fundamental barrier is rare and most exciting.
>
> 3)  The theorems are valid for any LSH irrespective of the choice of similarity, similar to why importance sampling is unbiased for any proposal.  So we don't really see what the issue is.
>
> 4) see 1
>
> 5) We will proofread the paper. Thanks for pointing out.
>
> We hope that our comments will change the opinion of the reviewer. We are happy to have any more suggestions.
> Thanks for the time in providing feedback.

---

### Decision · Program_Chairs · 2018-01-29
**ICLR 2018 Conference Acceptance Decision**

**Decision:**

Invite to Workshop Track

**Comment:**

The reviewers think that the theoretical contribution is not significant on its own. The reviewers find the empirical aspect of the paper interesting, but more analysis of the empirical behavior is required, especially for large datasets. Even for small datasets with input augmentation (e.g. random crops in CIFAR-10) the pre-processing can become prohibitive. I recommend improving the manuscript for a re-submission to another venue and an ICLR workshop presentation.